*Method*

# KCML: a machine-learning framework for inference of multi-scale gene functions from genetic perturbation screens

Heba Z Sailem[1,2,*] (iD), Jens Rittscher[1,2] (iD) & Lucas Pelkmans[3] (iD)

## Abstract

Characterising context-dependent gene functions is crucial for understanding the genetic bases of health and disease. To date, inference of gene functions from large-scale genetic perturbation screens is based on *ad hoc* analysis pipelines involving unsupervised clustering and functional enrichment. We present Knowledge- and Context-driven Machine Learning (KCML), a framework that systematically predicts multiple context-specific functions for a given gene based on the similarity of its perturbation phenotype to those with known function. As a proof of concept, we test KCML on three datasets describing phenotypes at the molecular, cellular and population levels and show that it outperforms traditional analysis pipelines. In particular, KCML identified an abnormal multicellular organisation phenotype associated with the depletion of olfactory receptors, and TGFβ and WNT signalling genes in colorectal cancer cells. We validate these predictions in colorectal cancer patients and show that olfactory receptors expression is predictive of worse patient outcomes. These results highlight KCML as a systematic framework for discovering novel scale-crossing and context-dependent gene functions. KCML is highly generalisable and applicable to various large-scale genetic perturbation screens.

**Keywords** cell morphology and microenvironment; CRISPR and siRNA screening; functional genomics; high content screening; olfactory receptors
**Subject Categories** Computational Biology; Methods & Resources
**Mol Syst Biol. (2020) 16: e9083**

## Introduction

Despite the deluge of acquired datasets with high-throughput gene perturbation screening (HT-GPS), the function of a large number of human genes remains poorly understood (Dey *et al*, 2015). Moreover, gene ontology (GO), the most comprehensive and structured annotation of gene functions, is largely limited to cell type- and context-independent gene functions (Huntley *et al*, 2015). However, gene function is highly contextual, even for unicellular organisms (Radivojac *et al*, 2013; Liberali *et al*, 2014). Therefore, there is an urgent need for new methods that allow data-driven and context-dependent functional gene discovery based on more complex phenotypes of multicellular organisms.

Although HT-GPS has proved to be a powerful method for discovering novel gene functions, the analysis of these datasets has remained a challenging task. This is due to the complexity of phenotypes that the perturbation of a single gene can lead to, as a gene can participate in different functions at different scales. These functions depend on the gene product localisation in the cell (e.g., cytoplasm versus nucleus for transcription factors), cell cycle state (e.g., G1, G2 or S phase), cell type, cell–cell and cell–microenvironment interactions and treatment conditions (Sero *et al*, 2015). Existing analysis pipelines based on unsupervised clustering do not generally account for these factors. Consequently, resulting phenotypic clusters are difficult to interpret as they might be composed of different subphenotypes (Yin *et al*, 2013; Sailem *et al*, 2014). These challenges are often avoided, particularly in image-based screens, by analysing only a small fraction of the information contained in HT-GPS datasets (Singh *et al*, 2014) which greatly underutilises their potential.

Supervised machine learning has been applied successfully in many HT-GPS studies (Held *et al*, 2010; Neumann *et al*, 2010; Shariff *et al*, 2010; Sullivan *et al*, 2018; Eraslan *et al*, 2019). One attractive solution for addressing the lack of phenotypic annotations is the utilisation of existing biological knowledge to build intelligent systems that can identify functionally relevant features and phenotypes. This approach is weakly supervised as existing knowledge can only provide noisy labels (Dutta *et al*, 2020). Approaches that utilise existing functional annotations have been successfully applied to inference of pathways activity (Schubert *et al*, 2018) as well as prediction of protein functions from multiple data types including protein sequence and structure, phylogeny, as well as protein–protein interactions and gene co-expression networks (Radivojac *et al*, 2013; Dey *et al*, 2015; Jiang *et al*, 2016).

1 Department of Engineering Science, Institute of Biomedical Engineering, University of Oxford, Oxford, UK
2 Big Data Institute, Li Ka Shing Centre for Health Information and Discovery, University of Oxford, Oxford, UK
3 Department of Molecular Life Sciences, University of Zurich, Zurich, Switzerland
*Corresponding author. Tel: +44 1865 617675; E-mail: heba.sailem@eng.ox.ac.uk

Additionally, pioneering work has been done in inferring data-driven gene ontology in yeast (Kramer *et al*, 2014; Yu *et al*, 2016; Ma *et al*, 2018). However, to our knowledge, this approach has not been applied in the context of large-scale HT-GPS datasets in multicellular organisms where genetic redundancy and phenotypic complexity are much higher.

Image-based screens are particularly advantageous for inference of biological functions as they provide spatial and context information at the single-cell level which allow capturing the emergent behaviours in biological systems (Lock & Strömblad, 2010). Single-cell data are critical for identifying loss-of-function phenotypes that are dependent on cellular state or manifest in only a small subpopulation of cells (partial penetrance) (Sacher *et al*, 2008). However, even for a widely used marker such as DAPI, phenotypic information on nuclear morphology and organisation of cells is often not fully utilised. The importance of studying the functional relevance of nuclear morphology and multicellular organisation is underlined by the fact that this information is successfully used by pathologists for patient diagnosis based on haematoxylin and eosin-stained tumour sections (He *et al*, 2012; Uhler & Shivashankar, 2018). Comprehensive analysis of changes in cell morphology and microenvironment following perturbation is crucial for the identification of genes associated with these important biological traits.

Systematic evaluation of gene sets in biological contexts different from the ones in which they are known to function can uncover valuable insights into the regulation of biological systems. For example, the roles of genes within the context of development, such as mesoderm development (MSD), which involves the coordination of cell migration, cell adhesion and cytoskeletal organisation through TGFβ and WNT signalling, are often deregulated within the context of colorectal cancer (Klinowska *et al*, 1994; McMahon *et al*, 2010; Kiecker *et al*, 2016). Therefore, identification of phenotypic signatures that are associated with perturbing MSD genes might shed light on how dysregulation of MSD genes can contribute to colorectal cancer development.

The importance of characterising context-dependent gene functions can be exemplified by the increasing evidence on the role of olfactory receptors in diseases such as cancer (Lee *et al*, 2019). Olfactory receptors constitute the largest gene family in humans (~400 genes) which were discovered in 1991 in sensory neurons. However, their functions in non-sensory tissues are poorly understood (Maßberg & Hatt, 2018). Investigating the similarity of olfactory receptor perturbation phenotypes to the perturbation of known gene programmes would enable the discovery of their functions in different tissue types.

Here, we propose KCML, a novel framework for automated knowledge discovery from large-scale HT-GPS. KCML is designed to account for pleiotropic and partially penetrant phenotypic effects of gene loss. We apply this framework to three large-scale datasets generated by different methods, describing phenotypes at the molecular, cellular and tissue levels, and show that it outperforms existing analysis pipelines. We analyse a cell organisation phenotype that KCML identifies and links to genes annotated to the Mesoderm Development (MSD) term. KCML predictions include many genes in TGFβ and WNT signalling pathways as well as many olfactory receptors. Through an integrative analysis with gene expression data of colorectal cancer patients, we validate the link between the expression of olfactory receptors and TGFβ and WNT signalling and show that the expression of some olfactory receptors can stratify the outcome of higher-grade colorectal cancer patients. In summary, KCML is a flexible and systematic framework for comprehensively analysing HT-GPS datasets and identifying context and tissue-dependent gene functions.

# Results

## A systematic framework for inferring gene functions from high-dimensional phenotypic data

KCML aims to utilise existing biological knowledge, as captured by GO, to automatically identify gene perturbation phenotypes in HT-GPS datasets and map these phenotypes to potential biological functions. Central to our approach are GO term classifiers that effectively link gene annotations, which are not specific to a cell type or biological context, to the rich contextual information provided in HT-GPS datasets. Critically, each term classifier identifies the phenotypic signature associated with a given gene annotation, and it can therefore be used to study how perturbation of biological functions at lower-scale contributes to higher-scale phenotypes.

Here, a weakly supervised learning approach is used to train each term classifier. Binary support vector machine (SVM) classifiers prove to be well suited to this problem (Materials and Methods). Each GO term classifier learns to discriminate between perturbation phenotypic profiles of genes annotated to that term (positive class) and a random set of remaining genes (negative class) (Fig 1A). To select features that are relevant to a given biological function, we initially train the GO term classifier using one feature and iteratively add features to the model if they improve the performance of the classification based on the F-score metric (i.e., forward feature selection; see Materials and Methods). Only those classifiers that exceed a certain performance threshold on unseen data are used for prediction (Materials and Methods). For example, a classifier of the GO term "cell cycle" will select features that are discriminative of the perturbation effect of the annotated cell cycle genes and define a decision boundary separating these effects from a random set of negative genes (Fig 1A). If such a classifier can successfully predict a held-out sample of annotated cell cycle genes, then it is used to predict other potential cell cycle genes based on phenotypic similarity. These predictions can be ranked based on the distance to the SVM decision boundary, which indicates the strength of the phenotype. Predictions from different GO term classifiers are then combined to yield a data-driven multi-ranking of gene functions. One of the main advantages of this approach is the effective augmentation of GO annotations in a context-dependent manner.

We train GO term classifiers from the three GO ontologies: biological process, molecular function and cellular component (Fig 1B, Materials and Methods and Table EV1). As GO terms can vary in specificity and number of annotated genes depending on their position in the GO tree, we only included terms with a sufficient number of positive examples (100–500 annotated genes). On average, each gene has 30 annotations (Fig 1C) with 2,152 genes having more than 50 annotations and 2,908 genes having no annotations based on the selected terms.

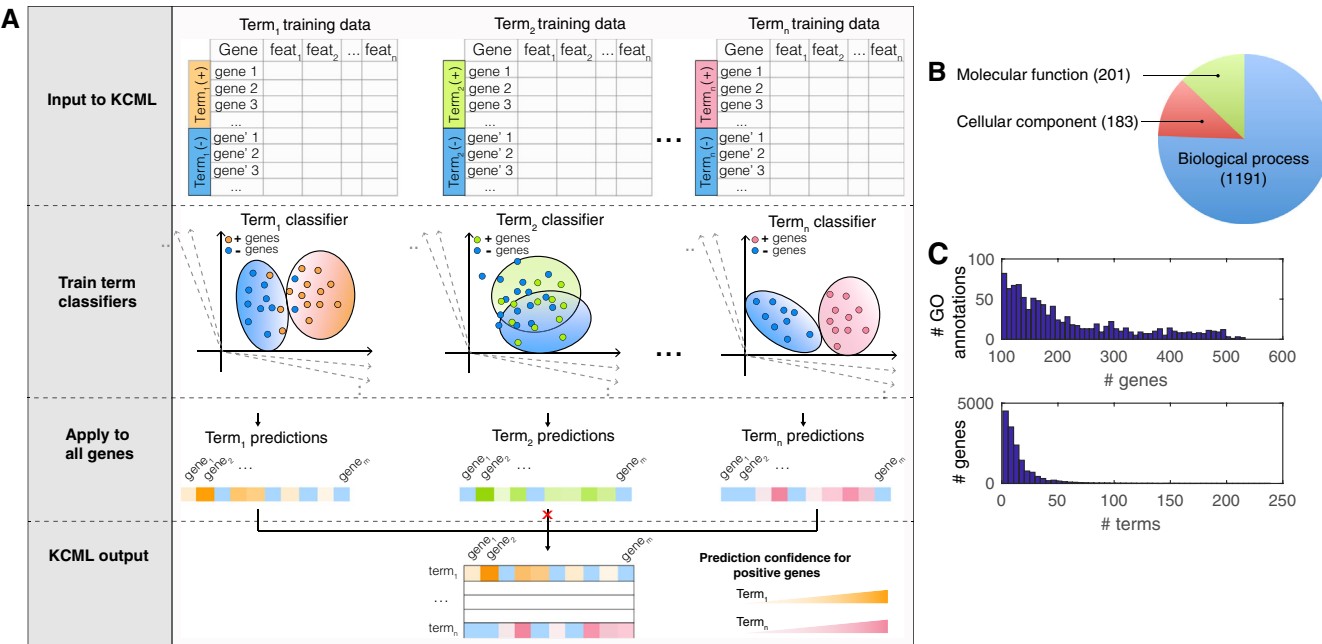

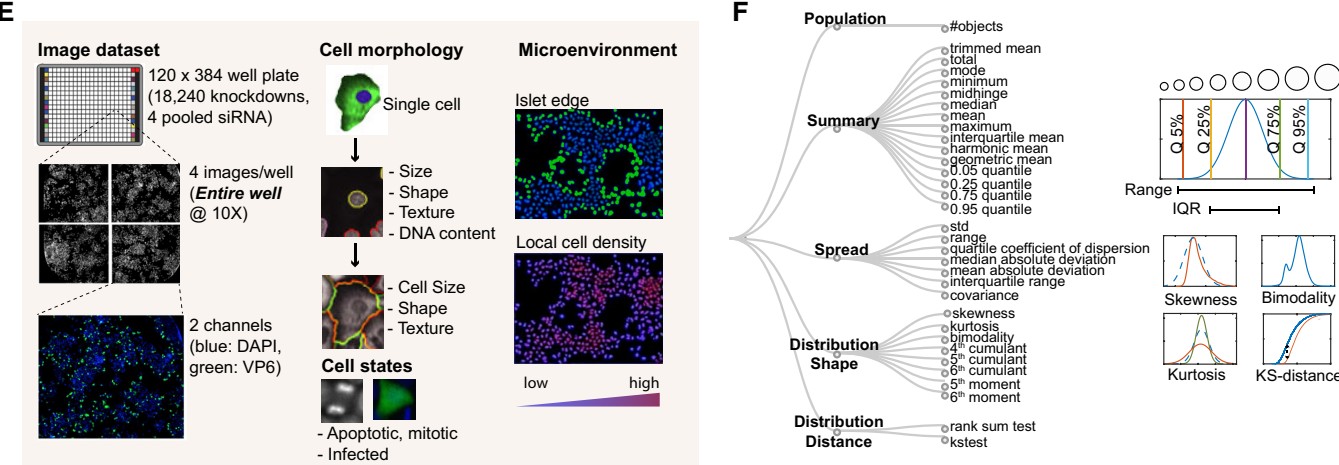

| | Expression | Image-based | Viability |
|---|---|---|---|
| **Scale** | Molecular level | Single cell >> population | Population-level |
| **Biological context** | breast cancer (MCF7) | colon cancer (HCT116) | 60 cell lines (12 tissues) |
| **Technology** | siRNA | siRNA | CRISPR |
| **#Perturbations** | 11,536 knockdowns | 18,240 knockdowns | 15,055 knockdowns |
| **#Features** | 3,287 features | 1,719 features | 80 features |
| **Phenotype** | bulk gene expression | morphology, microenvironment and infection | viability |

**Figure 1. KCML workflow for inferring gene functions from different large-scale genetic datasets.**

A   KCML workflow trains multiple classifiers to identify phenotypes and genes associated with different gene ontology terms. Feat: feature.

B   Categories and numbers of included GO terms.

C   The distribution of genes per term and vice versa based on GO annotations.

D   Tested datasets.

E, F   (E) Overview of single-cell-resolved image-based features measuring morphology, microenvironment and infection (F) and the generated statistics based on a population of single cells when applicable (*n* > 625 and Table EV2). Examples on the various measurement types are shown on the right and include multiple quantiles, mean, range, IQR (interquartile range), kurtosis, skewness, bimodality and KS distance.

## KCML generalisability

To illustrate the generalisability of KCML to various genetic perturbation screens, we applied it to three datasets measuring different types of phenotypic data and utilising different experimental technologies: (i) pooled genome-wide CRISPR/Cas9 screens measuring cell viability in 60 cancer cell lines (cell population-level phenotype) (Rauscher *et al*, 2018); (ii) a large-scale siRNA screen measuring changes in the expression of 3,287 genes in MCF7 breast cancer cells (transcriptome-level phenotype) (Duan *et al*, 2014); and (iii) an arrayed image-based genome-wide siRNA screen measuring changes in 168 single-cell features that quantify morphology, microenvironment and infection in HCT116 colorectal cancer cells based on stains of DAPI and the rotavirus-expressed viral protein 6 (VP6, single-cell and population-level phenotypes) (Green & Pelkmans, 2016) (Fig 1D and E). The latter dataset is composed of single-cell measurements, in which each perturbed population has 6,040 cells on average (Fig EV1A). To capture the heterogeneity in cellular states and collective cellular behaviour as readouts, we aggregated single-cell features into 1,719 features per gene perturbation profile. Computed statistical measures include standard deviation (SD), various quantiles, skewness, kurtosis, bimodality coefficient, Kolmogorov–Smirnov (KS) (Altschuler & Wu, 2010) and rank-sum statistics (Fig 1F, Materials and Methods and Table EV2). Furthermore, we corrected all features to detect cellular phenotypes independent of cell number (Fig EV1A–D and Materials and Methods).

## Performance and validation of KCML

For each of the datasets, we selected the most confident GO term classifiers based on the recall (true-positive rate) and false-positive rate such that it is significantly better than classifying a random set of genes (Fig 2A and Materials and Methods). The average area under the ROC curve (AUROC), which reflects specificity versus sensitivity, for the three datasets is 75.44% (Figs 2A and EV1E). The HT-GPS dataset using gene expression as a readout had the highest recall and AUROC, as well as the number of classifiable terms—i.e., classifiers scoring above our selection threshold (Fig 2A and B). This is expected since this dataset measures, for each perturbation, changes in the expression of 3,287 genes, whose variation is representative for most of the transcriptome (Duan *et al*, 2014). Interestingly, viability measured in 60 cancer cell lines from 12 tissue types can also be informative of gene functions as the number of classifiable terms is comparable with the multivariate image-based screen (Fig 2A and B).

Multivariate single-cell-resolved image-based readouts outperformed gene expression for some GO terms, while 33 terms were only classifiable based on image-based single-cell features (Fig EV1F and G). Many of those are involved in phosphorylation, ubiquitination or membrane transport. This might be due to the fact that bulk gene expression does not necessarily capture phenotypic effects caused by post-transcriptional events, or measure changes at the single-cell level. For example, exopeptidase activity and voltage-gated channels had a higher overall recall based on image-based single-cell features compared to the gene expression dataset (Fig EV1F). These results show that different experimental techniques for probing biological systems complement each other and provide different functional information.

The GO term classifiers obtained using KCML captured biologically relevant signatures, as they performed significantly better than classifiers based on random sets of genes (Fig 2C, $P < 1.8443e-17$ and Materials and Methods). Furthermore, KCML outperformed commonly used analysis pipelines for HT-GPS datasets based on dimensionality reduction and unsupervised clustering. Clustering of gene profiles using k-means and self-organising maps while varying the number of clusters resulted in AUROC of 50%, which indicates random classification (Fig 2D and Materials and Methods). Similar results were obtained even when we considered pairwise correlation between genes in a given GO term (Fig 2D and Materials and Methods). These results confirm the power of KCML in identifying functionally relevant phenotypes and highlight the need for such a systematic approach, as existing analysis methods do not provide a clear functional interpretation of quantitative phenotypes.

To obtain unbiased validation of the KCML predictions, which were based on annotations downloaded from UniProt in March 2018 (annotations$_{2018}$), we downloaded annotations accumulated in UniProt in July 2019 (annotations$_{2019}$) and used the new annotations between March 2018 and July 2019 (annotations$_{2019}$-annotations$_{2018}$) as a validation set (11,383 annotations) (Fig 2E). Since these new annotations were never seen by KCML, and were based on diverse resources, they are not likely biased to a particular dataset. We found that KCML can significantly predict many of the newly reported annotations with comparable performance for the different datasets (Fisher's exact test right-tail $P < 0.025e-11$ and Fig 2F). Furthermore, the performance of our models on annotations$_{2018}$ correlates with the performance on the new UniProt annotations (Fig 2G). Therefore, GO term classifier performance can be considered for selecting the more likely hypotheses.

An example of a gene for which new UniProt annotations are consistent with KCML predictions is URB1; a homolog of the

---

**Figure 2. KCML performs best in gene function inference and predicts many novel gene functions.**

A    AUROC and recall of test samples of classifiable terms across the three tested datasets ($n > 141$).

B    The number of classifiable terms in the three datasets.

C, D  Comparison of KCML against classifiers that were trained to classify random sets of 200 genes ($P < 1.8e-17$) (C), or clustering of genes using k-means, SOM (self-organising maps) or enrichment for pairwise correlations ($P < 1e-150$) (D) ($n > 100$).

E    Validation annotations are defined based on annotations accumulated between Mar 2018 and July 2019.

F    Percentage of new GO annotations that was predicted by KCML correctly.

G    Correlation between the performance of GO term classifiers on training annotations (annotations$_{2018}$) versus validation annotations (annotations$_{2019-2018}$).

H    Predictions of many GO terms are significantly enriched for first neighbour interactions based on STRING, physical (based on experimental evidence in STRING) and Pathway Commons databases (hypergeometric test). Error bars indicate 1 standard deviation ($n = 145, 141, 688$ from left to right).

Data information: Box plots elements: centre line, median; box limits, 25$^{th}$ and 75$^{th}$ percentiles; whiskers, $\pm$ 2.7 standard deviation. Points: outliers.

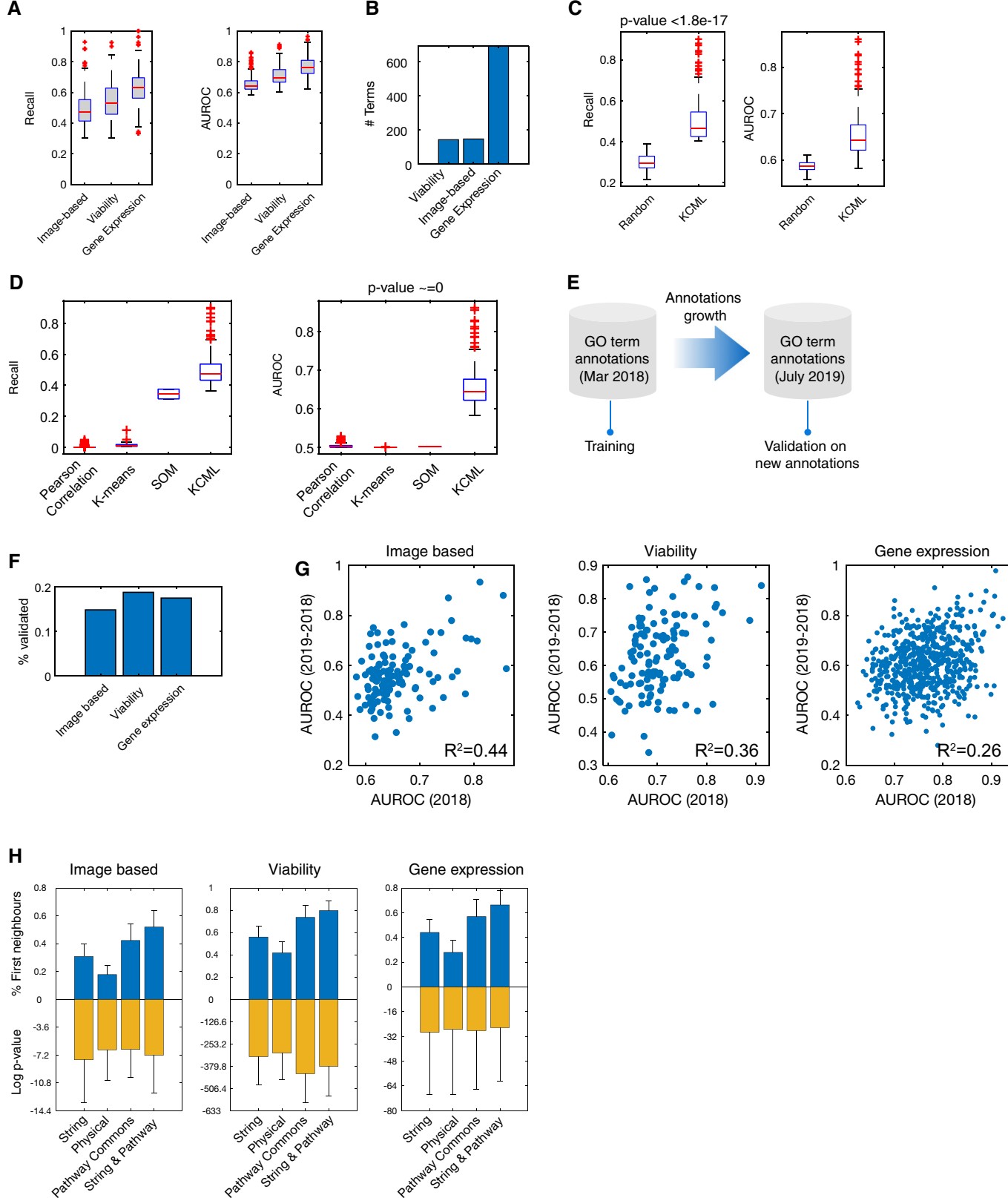

**Figure 2.**

essential Npa1p yeast gene. URB1 has been characterised to play a role in early steps of 60S ribosomal subunit biogenesis via associating with RNP proteins, and it localises to the fibrillar centre of the nucleolus, where ribosome biogenesis takes place (Uhlén *et al*, 2015; Farley-Barnes *et al*, 2018). In August 2018, UniProt had assigned URB1 a new function: non-coding RNA processing based on a phylogeny analysis. KCML predicted that URB1 is involved in non-coding RNA processing from each of the three datasets analysed, which were generated by different laboratories and different gene perturbation technologies. URB1 interacts with 26 ncRNA-processing proteins based on STRING and Pathway Commons databases, which further supports this prediction.

Other examples of new UniProt annotations that are consistent with KCML predictions include the following: (i) All members of CD1 glycoprotein family (CD1A, CD1B, CD1C, CD1D and CD1E) were correctly predicted to be associated with "regulation of the adaptive immune response" based on the siRNA screen with transcriptome readout; and (ii) the mini-chromosome maintenance (MCM) complex proteins (MCM2-7), which are known to be involved in DNA replication, were correctly predicted to also be involved in double-strand break repair, based on both the CRISPR/Cas9 cell population viability screens and the siRNA screen with transcriptome readout. These examples illustrate the power and generalisability of KCML in predicting new gene functions based on gene perturbation screens which can be useful for generating data-driven hypotheses.

As gene ontology annotations are noisy and incomplete, KCML can generate a high number of potential hypotheses. We sought to determine whether these predictions can be due to the propagation of perturbation effects to neighbouring genes in the protein–protein interaction network. Especially, it is expected that interacting genes are more likely to be functionally related and their depletion to result in a similar phenotype (Evans *et al*, 2013). We found that a large proportion of predicted genes for a given GO term do interact with the genes that are previously annotated to that term, based on STRING and Pathway Commons databases (Fig 2H). This enrichment is significant for many of the terms. Moreover, accounting for protein–protein interactions results in a large improvement in precision–recall curves (Fig EV1H). This provides further validation of KCML predictions, where neighbouring genes in the interaction network are more likely to perform similar functions.

### Using KCML to identify GO terms represented in the image-based dataset and their relationships

To demonstrate the use of KCML, we focus on the further analysis of the image-based perturbation dataset, as it is the only dataset that

provides spatially resolved single-cell measurements, which are particularly challenging to analyse and interpret (Collins, 2009). We trained KCML using a subset of features to determine the GO terms that can be learned from different cellular markers (morphology and microenvironment features based on DAPI versus infection features based on VP6 staining) (Materials and Methods). Shape features are predictive of 61 GO terms while infection measurements are predictive of 38 terms, with 8 terms shared (Figs 3A–C and EV2A). Combining both feature sets result in an additional 54 classifiable terms (Fig 3A). These results illustrate that multivariate imaging data can be predictive of many gene functions even when only one or two cellular stains are used.

To gain insight into the functions that have been learned by KCML, we generated a network of classifiable GO terms based on the overlap in their predicted gene lists (Figs 3D and EV2B and C). We observe a strong cluster of membrane transport-related terms including potassium, calcium, sodium and metal ion channels (Fig 3D). Most of these terms can be classified using morphology and microenvironment features alone and their phenotypic profiles cluster together (Figs 4A and EV3). Interestingly, based on our data, ion channel terms (molecular-level) are linked to multicellular organismal signalling function (tissue level). Another cluster included many phosphorylation- and ubiquitination-related terms, many of which are classifiable based on infection features (Fig 3D). Thus, KCML can classify biological functions that act at different scales, from the molecular to the tissue level.

### KCML automatically maps GO terms to functional phenotypes

To understand the phenotypic changes associated with different GO terms, we categorised the phenotypic features based on feature type (cell context, morphology, DAPI intensity and texture, VP6 intensity and texture, state, etc.) as well as measurement type (summary statistics, spread statistics, distribution shape or distribution distance—the distance between perturbed and control distributions) (Fig EV2E and Table EV2). We then counted and scaled the number of features selected by a term classifier in the different categories (Figs 4B and EV2D). Notably, the rank-sum statistic was among the most selected measure by different classifiers indicating its biological relevance and robustness (Fig EV2E).

Membrane transport-related terms are predicted based on the increased number of cells and total cell area as well as local cell density and cell distance to islet edge (Figs 4B and EV3). Surprisingly, the "ligand gated ion channel activity" and "voltage-gated ion channel activity" terms are the most accurately classified (Fig EV2A). Depletion of chloride transport genes also affected many texture and intensity measures of VP6, but most interestingly, it significantly decreased rotavirus infection ($P < 4.9e-139$, Figs 4B

---

**Figure 3. Analysis of functional information using KCML based on image-based dataset.**

A   The number of classifiable terms when using morphology versus infection features.

B   AUROC and recall of test samples for classifiable terms when using different subsets of features ($n > 38$). Box plots elements: centre line, median; box limits, 25th and 75th percentiles; whiskers, ± 2.7 standard deviation. Points: outliers.

C   The improvement in performance when all the features are used for terms that are classifiable when only shape features are used ($n = 61$), only infection features are used ($n = 38$), or either shape or infection features are used ($n = 8$). Only a slight improvement in classification performance is achieved by combining feature subsets when the signatures based on a subset of features are already strong. Error bars represent mean + SD.

D   Network representation of classifiable terms where edges indicate the overlap in the predicted gene lists between GO terms.

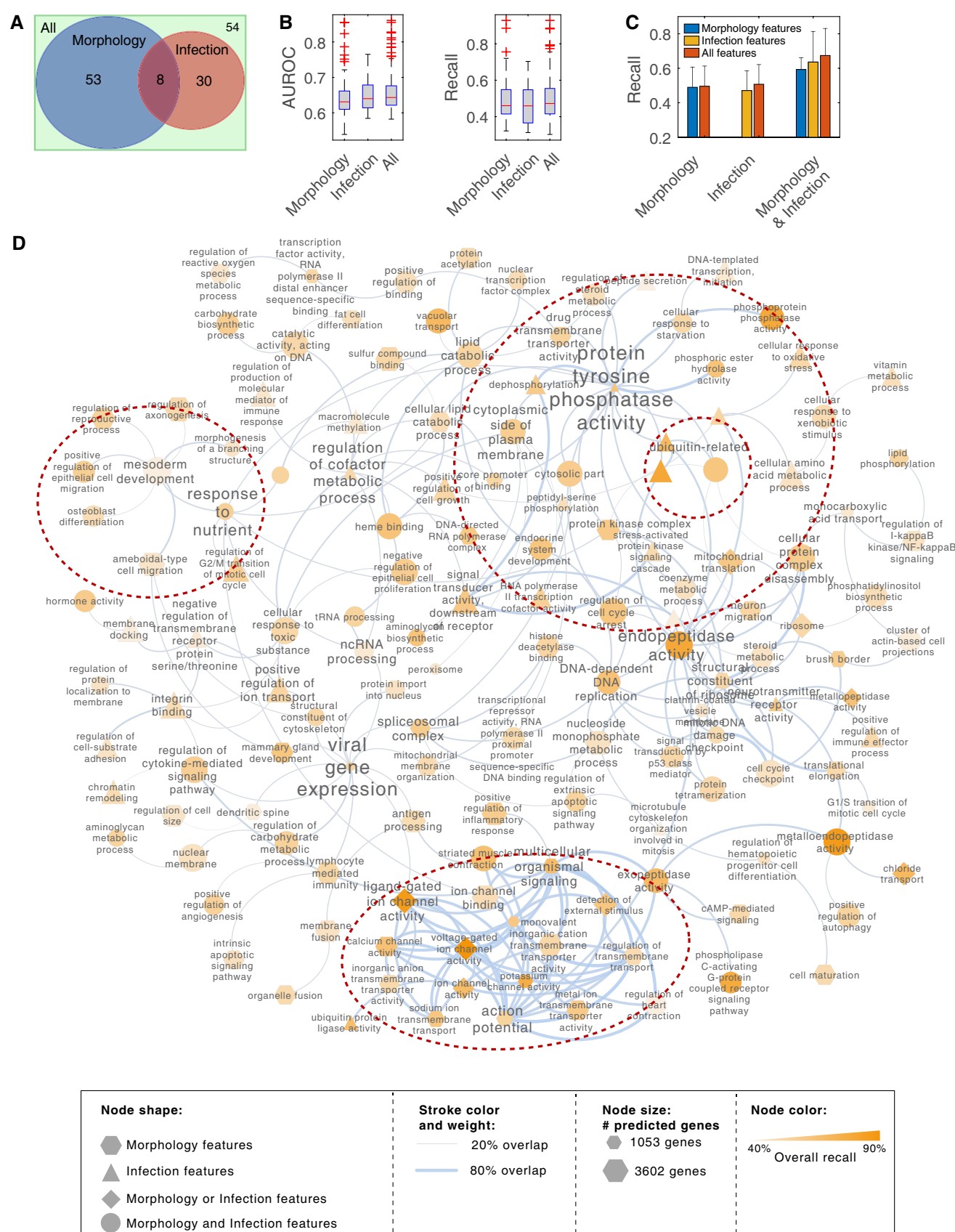

**Figure 3.**

and EV3A). These results illustrate the functional relevance of single-cell aggregated measurements of cell morphology and context and suggest that chloride channels might be required for the spread or replication of rotavirus.

From image-based data, KCML predicted a number of genes involved in mesoderm development (MSD). This was based on the spread and summary values of cell context, DAPI intensities, as well as the distribution shape of cell morphology, which are indicative of changes in cell organisation (Figs 4B and EV4A). Indeed, depletion of previously annotated (e.g., *SMAD3*) and predicted (e.g., *ITGAV*, *WNT8B* and *OR51B4*) MSD genes results in small cell colonies compared to control (Fig 4C). This phenotype might indicate the inability of cells to spread or migrate. This hypothesis is supported by the high overlap of the MSD-predicted genes with GO terms such as migration (epithelial and ameboidal-like) and morphogenesis of branching structure (Figs 3D and EV4A). Thus, through a holistic analysis of multiple features, KCML reveals a role for MSD-associated genes in the spread and organisation of epithelial colorectal cancer cells into a uniform epithelial sheet.

## KCML allows multi-scale analysis of high-dimensional data

Because KCML predicts GO terms associated with different biological scales, we asked whether it can subdivide higher-scale properties shared by a set of genes into gene subsets that carry out different aspects of the higher-scale property at lower scales. For example, previously annotated genes to a higher-scale GO term such as MSD are predicted by KCML to perform different functions at lower scales, such as positive regulation of epithelial cell migration and mammary gland development (Fig 5A). Interestingly, MSD genes that are predicted to play different functions at lower scales tend to occupy different regions in the MSD subphenotypic space, reflected by differences in a subset of MSD features (Figs 5B and EV4B). Thus, KCML allows specifying a common higher-scale phenotypic property emerging from the collective action of a set of genes into different lower-scale aspects of this property performed by subsets of these genes.

In addition, perturbation of a biological function often affects only a subset of the measured features. For example, the mean and integrated intensity of DAPI is significantly lower in genes predicted to participate in cell cycle checkpoint (Fig EV5A). However, these genes do not cluster together when all measured phenotypic features are considered, but only when considering features selected by the cell cycle checkpoint classifier (Fig EV5B and C). Importantly, identification of subphenotypic effects associated with each gene allows determination of which of those subphenotypic effects are potential off-target effects. Specifically, predictions of genes that

are targeted by the same siRNA seed (guide strand) are filtered out if the seed is significantly over-represented in a given GO term classifier (Fig EV5D–F and Materials and Methods). This emphasises the importance of searching in the subphenotypic space to deconvolve biological signals contained in high-dimensional data and obviate off-target effects.

## Validation of mesoderm development classifier predictions in the context of colorectal cancer

Since loss of healthy tissue organisation is one of the main characteristics in tumours (Hinck & Näthke, 2014), we sought to analyse MSD genes whose perturbation resulted in abnormal cell organisation between HCT116 cells in culture. As expected, MSD genes are significantly enriched for morphology-regulating pathways including tight junctions, focal adhesion and actin cytoskeleton (Figs 6A and EV6 and Table EV3). Among the predicted genes are 15 collagen, 12 integrin and many polarity genes, such as Par3 and Par6 (Table EV3). Moreover, many of the predicted MSD genes also participate in pathways that are often dysregulated in colorectal cancer including TGFβ, WNT and PI3K-AKT pathways (Muzny *et al*, 2012) (Fig 6A). Predicted genes include *TGFβR2*, *PTEN* and *ERBB2* which are often mutated in cancer (Kuipers *et al*, 2015). TGFβ and WNT signalling are known to contribute to mesoderm development and their over-activation is associated with mesenchymal and stemness phenotypes in colorectal cancer, respectively (Hinck & Näthke, 2014). Collectively, these results illustrate how KCML allows constructing an integrative view of how modular gene programmes coordinate different signalling pathways to drive cellular phenotypes.

As known MSD genes are highly implicated in colorectal cancer, we sought to determine the relevance of predicted MSD genes in colorectal cancer patients. We interrogated The Cancer Genome Atlas (TCGA) gene expression dataset (Muzny *et al*, 2012) of 577 colorectal cancer patients. Four consensus molecular subtypes (CMS) of colorectal cancer have been identified: CMS1 (microsatellite instability), CMS2 (WNT activation), CMS3 (metabolic) and CMS4 (mesenchymal) (Guinney *et al*, 2015). Strikingly, the top 300-predicted MSD genes recapitulate colorectal cancer molecular subtypes with comparable performance to using all genes or known mesoderm genes (Fig 6C, Materials and Methods). This is significantly higher than random sets of 300 genes ($P \leq 3.44e\text{-}25$ and Materials and Methods). Therefore, MSD genes that alter epithelial cell organisation in HCT116 cells can stratify colorectal cancer patient's molecular subtypes.

Next, we sought to validate the relationship between TGFβ and WNT signalling, and expression values of predicted MSD genes in colorectal cancer. Average expression of WNT genes is used as a

---

**Figure 4. Association between phenotypic changes in single-cell-based dataset and GO terms.**

A Hierarchical clustering of the average fold change of predicted positive versus negative samples for each term using all features. Hierarchical clustering is based on ward linkage and Euclidean distance. Regions outlined by cyan rectangles are shown in Figs EV3 and EV4A.

B The number of features in different categories that are selected by the respective GO term (scaled). Blue indicates the number of features with a higher average than control, while red indicates the number of features with a lower average than control. * indicates the GO terms that are classifiable by either shape or infection features.

C Example cell images following knockdown (k/d) of known or predicted MSD genes versus control (scrambled). (ii) zoom-in image of the region highlighted in (i). Blue: DAPI, and green: VP6 antibody. Scale bars = 65 μm.

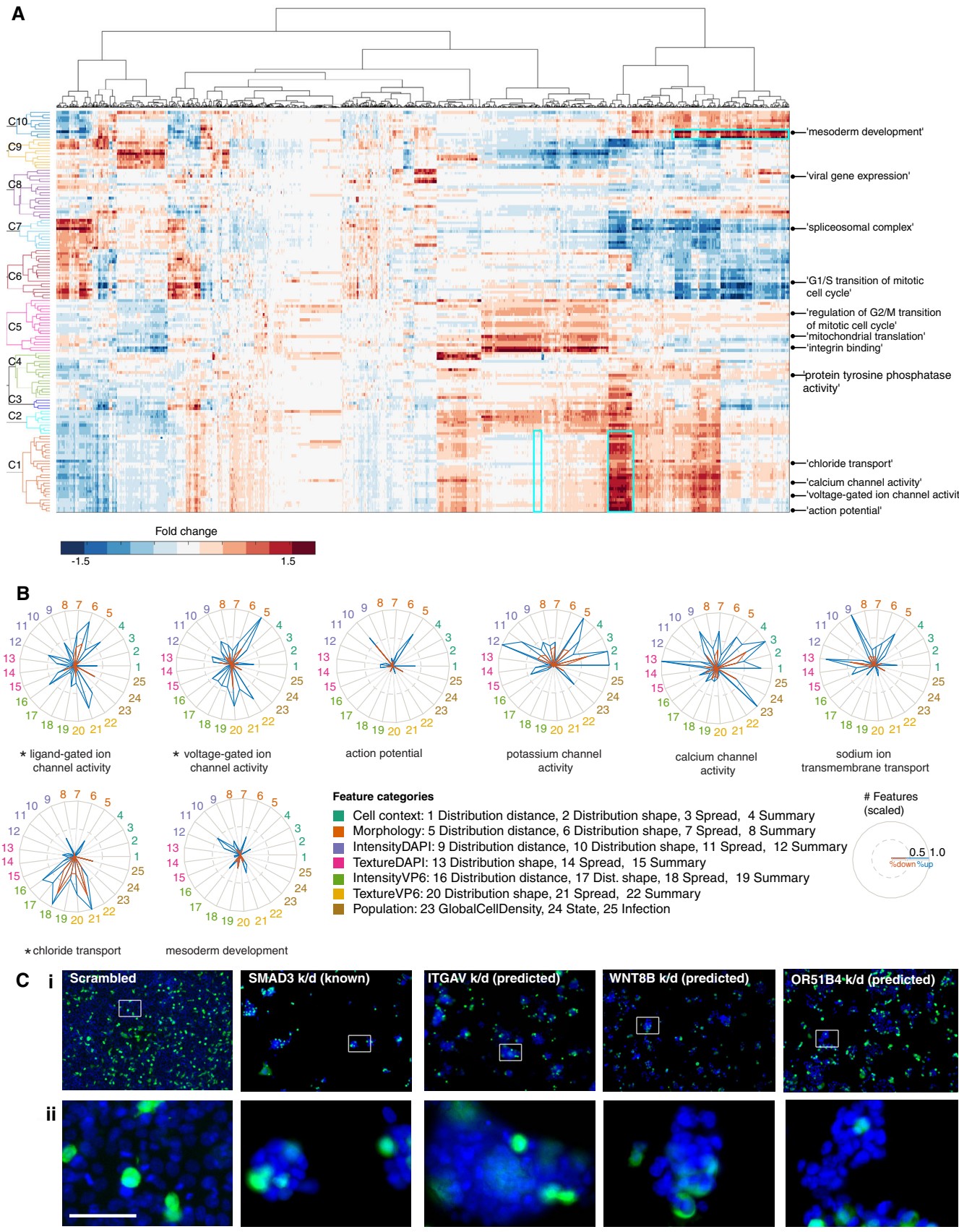

Figure 4.

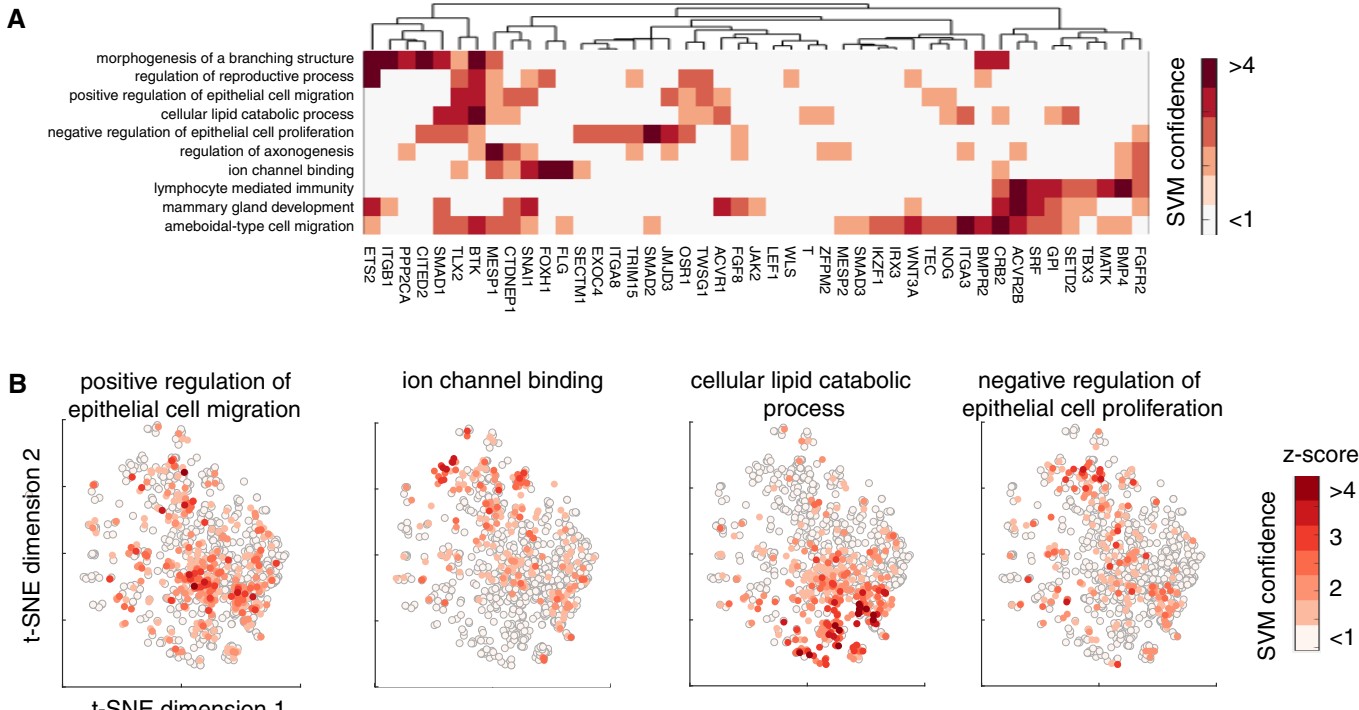

**Figure 5. KCML allows pleiotropic analysis of high-dimensional phenotypic data.**

A Heatmap showing SVM-based ranks (z-scored) for known MSD genes against multiple functions. The functions on the y-axis are the top ten overlapping terms with MSD classifier prediction.

B Embedding of MSD subphenotypic space using t-SNE based on the selected features by MSD classifier where only MSD genes are considered (Materials and Methods). Colour indicates the respective SVM rank for each gene for the corresponding function.

surrogate for WNT signalling while the TGFβ signature is based on genes reported by a previous study (Calon *et al*, 2012) (Materials and Methods). We found a significant correlation between MSD genes and TGFβ and WNT signatures in colorectal cancer patients (Fig 6D). The correlation between the predicted MSD genes and TGFβ/WNT signalling in colorectal cancer patients further supports their functional interaction.

Strikingly, 83 olfactory receptors were predicted by KCML to be involved in MSD, and their average expression correlates with WNT and TGFβ signatures (Fig 6B and D, Table EV4). These include

OR51B4 and OR5K1 genes that are over-expressed in HCT116 cells when compared to 947 cancer cell lines (Barretina *et al*, 2012). We sought to investigate the correlation between olfactory receptors and colorectal cancer patient outcomes as their application in colorectal cancer therapeutics is beginning to emerge (Lee *et al*, 2019). OR51B4 and OR5K1 are mostly expressed in patients with tumour grade 3 or higher (Fig 6E and F). Their expression, albeit at a low level, is predictive of significantly worse patient outcome based on Kaplan–Meier log-rank test especially in grade 3 and 4 tumours (Figs 6G and H, and EV7A–E). Similar results are obtained when we

**Figure 6. MSD predicated gene list is associated with key colorectal cancer pathways and patient outcome.**

A KEGG pathways that are significantly represented in MSD genes (P < 0.05 based on right-tail Fisher's exact test). Previously annotated genes (known) to MSD are shown in red while others are predicted by KCML based on phenotypic similarity to known MSD genes.

B Network depicting the interactions between MSD genes based on STRING and Pathway Commons.

C TCGA colorectal cancer patients' data projected into the first two principal components based on the expression of all genes (left), and expression of the top 300 predicted MSD genes (middle) and the expression of known mesoderm genes.

D Spearman correlation coefficient between TGFβ or WNT signatures and the average of known MSD genes, average of predicted MSD genes (rank 1–300, rank 301–600 and rank 301–1,000) or average of MSD-associated olfactory receptors (P < 0.05). ORs: olfactory receptors.

E, F Survival of colorectal cancer patients (months) against the expression of *OR51B4* (E) and *OR5K1* (F). Colour indicates tumour grade where grade 1 is well differentiated, grade 2 is moderately differentiated, and grade 3 or 4 is poorly differentiated.

G–I Kaplan–Meier survival analysis of colorectal cancer patients based on Grade 3 ± tumours and the expression state of *OR51B4* (G), *OR5K1* (H) and olfactory receptor metagene (I), which is aggregated, based on the expression of many olfactory receptors (Materials and Methods and Fig EV7G).

J, K Wald statistic value based on Cox proportional hazard regression analysis of *OR5K1* predictivity of survival against tumour grade, presence of lymph nodes and metastasis. Detailed results for all tested variables and models significance are shown in Tables EV5–EV7. * indicates P < 0.05, and ** indicates P < 0.001.

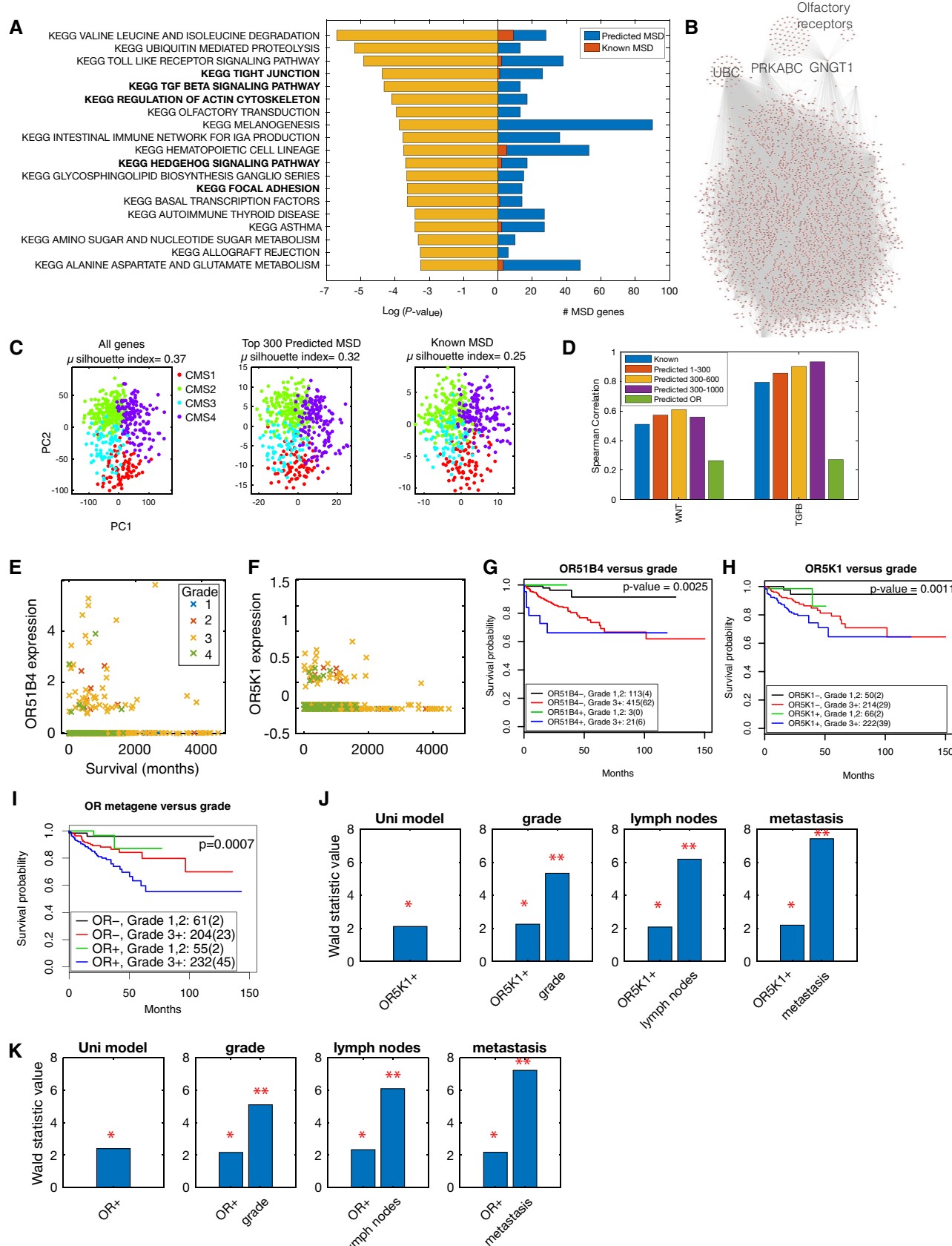

Figure 6.

aggregated the state of many MSD-associated olfactory receptors (Materials and Methods and Figs 6I and EV7F and G). This significant association was not observed using randomly selected genes (Materials and Methods). Importantly, expression of *OR5K1* or the olfactory receptor metagene is predictive of survival, independent of other clinical variables including tumour grade, presence of lymph nodes and metastasis state, as well as the expression levels of neighbouring oncogenes (Fig 6J and K, Tables EV5–EV7 and Materials and Methods). The expression values of these genes are unlikely to be due to misalignment with other olfactory receptors as they generally share less than 95% sequence similarity with other olfactory receptors (Fig EV7H and Table EV8). These findings confirm the association between expression values of olfactory genes and the WNT and TGFβ pathways, which is consistent with our KCML prediction based on gene perturbation phenotypes.

## Discussion

Intelligent machine-learning methods that systematically integrate existing biological knowledge are crucial for accommodating the explosive growth of phenotypic data at multiple system levels. Here, we propose a computable and flexible framework that integrates functional annotations to automatically identify three-way relationships between phenotypes, genes and biological functions. Not only does KCML outperform clustering-based approaches, but it also accounts for the pleiotropic nature of gene function and can mitigate the problem of off-target effects. We show that KCML generalises to different types of data and predicts novel gene functions based on various multivariate phenotypic readouts from images and transcriptome profiles.

A general problem in inferring functions from HT-GPS datasets is that the gene perturbation phenotype might be a result of affecting a biological function directly or indirectly through regulating a related function, which are propagated through protein–protein interaction networks. Indeed, we observe a high enrichment of protein–protein interactions for genes that are predicted to share a given term. This may explain the high number of positive predictions, as many neighbours in the interaction network are predicted to contribute to the same GO term. Another reason for the high number of predicted genes is that existing phenotypic profiling techniques only provide partial information on the phenotypic state of the cell and, therefore, more genes are likely to share this partial state. Advances in multi-omics approaches hold the promise to obtain more complete phenotypic profiles. Importantly, KCML allows prioritising predictions based on SVM confidence or against other biological annotations such as protein–protein interactions or KEGG pathways. Alternatively, high confidence predictions can be obtained by integrating results from various datasets. Taken together, our predictive analysis framework serves as a tool to generate testable hypotheses and pave the way to more integrative studies.

One limitation of our approach is the dependency on noisy GO annotations which do not provide perfect ground truth. Moreover, a sufficient number of positive genes are required to classify a certain biological function. Both of these factors can restrict the application of data-hungry deep learning methods and might influence KCML performance. Nonetheless, these issues also apply to existing pipelines and will be reduced as our databases of gene functions expand.

Cellular morphology has been illustrated to reflect multiple aspects of cell physiology (Fuchs *et al*, 2010; Sailem *et al*, 2014). Here, we further show that advanced measures of single-cell distributions in image-based screens are useful for identifying genes regulating multicellular functions. KCML identified that depletion of annotated ligand- or voltage-gated ion channel genes affects cell area and microenvironment measures at the population level. This is consistent with previous reports on the role of ion channels in cell volume regulation and proliferation (Lang *et al*, 2007). At a higher system level, KCML predictions for ion channel terms overlapped with multicellular organismal signalling. This implicates that ion channel perturbations affected communication between epithelial cells akin to their function in neurons (Barshack *et al*, 2008; House *et al*, 2015; Whited & Levin, 2019), but further experiments are required to validate that. Importantly, aggregated measures of single-cell features outperformed gene expression data in the identification of many membrane transport functions.

Our systematic analysis revealed a role for MSD genes in multicellular organisation, which manifested in cell clumping and increased local cell density in colorectal cancer cells. Cell microenvironment has been shown to contribute to cancer initiation and progression (Friedl & Alexander, 2011). Our analyses demonstrate that perturbation of many extracellular matrix and integrin genes can result in a similar phenotype to perturbing TGFβ and WNT signalling. This suggests an important link between cell microenvironment (adhesion) and shape (cytoskeleton) and determination of cell fate (i.e., stemness and differentiation via TGFβ and WNT signalling). Consistent with this, SMAD3, which plays an essential role in TGFβ signalling, has been shown to link shape information to transcription in breast cancer cells (Sailem & Bakal, 2017), while WNT signalling can be linked to cell microenvironment via the differential localisation of its downstream effector β-catenin. These results exemplify how KCML can be used to automatically interrogate quantitative phenotypic profiles to identify combinatorial use of modular gene programmes in different contexts.

The predicted role of olfactory receptors in MSD is consistent with previous reports on their expression in developing mesoderm tissue and their role in patterning (Dreyer, 2002; Nef & Nef, 2002; Weber *et al*, 2002). Moreover, they are associated with TGFβ and non-canonical WNT signalling in neuronal cells (Getchell *et al*, 2002; Zaghetto *et al*, 2007). There is increased evidence for the therapeutic potential role of olfactory receptors in many diseases including cancer (Lee *et al*, 2019). We show that the expression of *OR51B4* and *OR5K1* correlates with patient grade and worse prognosis. This supports the role of olfactory receptors in cell organisation, as higher-grade tumours are characterised by poorly differentiated cells and loss of epithelial tissue structure (Guinney *et al*, 2015). Olfactory receptors in the gut might be activated by odours and chemicals produced by microbes or ingested food. This might, in turn, activate dedifferentiation via crosstalk with TGFβ and WNT pathways. Thus, while more work is needed to investigate the underlying mechanisms, these results suggest that olfactory receptors might provide a potential biomarker for higher-grade colorectal cancer.

In summary, our results illustrate the generalisability and utility of KCML as a framework for systematic gene function discovery from HT-GPS datasets across multiple biological scales. We believe

that KCML can scale-up to more complex phenotypic screens probing micro-tissues and organisms or utilising advanced multiplexing technologies (Gut *et al*, 2018) and single-cell RNA sequencing (Norman *et al*, 2019). We envision that systematic application of

KCML to the large amounts of generated HT-GPS datasets will greatly accelerate and advance our understanding of gene functions at the molecular, cellular and tissue levels which can lead to the discovery of new therapeutic gene targets.

# Materials and Methods

## Reagents and Tools table

| Reagent/Resource | Reference or source | Identifier or catalog number |
|---|---|---|
| **Experimental models** | | |
| HCT116 (*H. sapiens*) | (Green & Pelkmans, 2016) | |
| MCF7 (*H. sapiens*) | (Duan *et al*, 2014) | |
| 60 cancer cell lines (*H. sapiens*) | (Rauscher *et al*, 2018) | |
| **Software** | | |
| Cytoscape v3.3.0 | http://www.cytoscape.org (Shannon *et al*, 2003) | |
| MATLAB 2019a | https://uk.mathworks.com | |
| CellProfiler | https://cellprofiler.org (Carpenter *et al*, 2006) | |
| R 3.6.2 | https://www.r-project.org | |
| **Other** | | |
| Gene ontology annotations | http://geneontology.org/docs/download-go-annotations/ | |
| Protein–protein interactions | https://www.string-db.org (Szklarczyk *et al*, 2017) and https://www.pathwaycommons.org (Cerami *et al*, 2011) | |
| TCGA colorectal cancer data | https://portal.gdc.cancer.gov (Muzny *et al*, 2012) | |
| Viability data | (Rauscher *et al*, 2018) | |
| Expression data of MCF7 | http://www.lincsproject.org/ (Duan *et al*, 2014) | |
| Sequence similarity of olfactory receptors | https://genome.weizmann.ac.il/horde/ (Olender *et al*, 2013) | |

### Methods and Protocols

#### KCML implementation and data analysis
KCML pipeline and all analyses were performed using MATLAB (http://www.mathworks.com/) unless stated otherwise.

#### Preparation of GO annotations
GO annotations were downloaded from UniProt (March 2018). Annotations of child terms were escalated up to parent terms in order to have a sufficient number of positive examples for classification. Only GO terms that had 100–500 gene annotations were considered resulting in 1,575 terms. Highly redundant GO terms (Jaccard index > 70%) were merged.

#### Gene profiles computation for image-based readouts
Image analysis was performed using CellProfiler as previously described (Green & Pelkmans, 2016). Briefly, all images were pre-processed using illumination correction and background subtraction. Nuclei were segmented based on DAPI channel. Cells were defined as 10-pixel expansion of the nuclei. Shape, intensity and texture features were quantified for DAPI and VP6 channels. SVM classifiers were trained to identify mitotic, apoptotic, poorly segmented and infected cells. Infection index was corrected for

population context (Green & Pelkmans, 2016). The number of cells in different states was normalised to the number of cells in the well.

Single-cell data were aggregated per well using various statistical measures (Table EV2). KS and rank-sum statistics were used to compare a treated population to scrambled/empty populations (distributions distance).

The rank-sum statistic was computed based on Mann–Whitney–Wilcoxon rank-sum test using ranksum MATLAB function. All values of a feature X were ranked when combining perturbed and control samples. The sum of the ranks of the perturbed sample was then used to calculate the statistic as following

$$U = R - \frac{n_1(n_1 + 1)}{2}$$

where $U$ is the rank-sum statistic, $R$ is the sum of cell ranks in the perturbed sample of cells, and $n_1$ is the size of the perturbed sample.

KS distance was computed as the KS statistic based on a two-sample Kolmogorov–Smirnov test using kstest2 MatLab function. KS distance measures the maximum distance between the cumulative distribution of perturbed and control cell populations.

Both KS statistic and rank-sum test statistic were multiplied by a scaling factor sf to account for differences in sample sizes.

$$sf = \sqrt{\frac{n_1 \times n_2}{n_1 + n_2}}$$

where $n_1$ is the size of the perturbed sample, and $n_2$ is the size of the control sample.

### Feature subsets of image-based dataset

All features based on DAPI channel as well as cell context measures were defined as morphology features. On the other hand, features based on VP6 channel including texture and intensity were defined as infection features. These also include the microenvironment of the infected cells such as the number of infected cells on the islet edge.

### Data pre-processing

All datasets were cleaned by excluding samples or features with more than 30% missing data. Remaining missing values were imputed based on the weighted mean of the nearest 10 neighbouring features based on Euclidean distance.

1   *siRNA gene expression:* siRNA gene expression profiles were averaged per gene. Principal component analysis was performed, and the first 1,000 principal components (*z*-scored) were used for KCML classification.
2   *Viability: z*-score was used to normalise the data.
3   *Image-based datasets:* Genes that significantly reduced viability (< 625 cells) were filtered to avoid SVM bias towards viability phenotypes (Green & Pelkmans, 2016). All data were *z*-scored by subtracting the plate mean and dividing by the plate standard deviation (SD). Then, the average per gene perturbation was computed. To eliminate features dependency on cell number, cell number was binned into 32 bins such that each bin had at least 100 gene perturbation profiles. Finally, feature values for samples in each bin were *z*-scored to the corresponding bin mean and SD.

### KCML training

Training of classification models: The same training pipeline was applied to the three datasets. The annotated genes for each GO term were split into 70% for training and 30% for testing. Different classification methods were tested on a subset of terms including Random Forests, Lasso Regression and SVM. SVM was chosen because it resulted in less overfitting (results are not shown). A binary SVM with a Radial Basis Function (RBF) kernel was trained per term to classify the annotated genes (positive class) against a set of randomly selected genes excluding annotated genes (negative class). SVM was trained based on a balanced number of genes in the negative and positive class. All training was performed with a 30-fold cross-validation for training samples to avoid overfitting. Scale of SVM kernel (sigma) was optimised by brute force search based on *F*-score metric. Different sigma values were tested for each dataset. *F*-score consider sensitivity as well as specificity of the model by computing the harmonic mean of precision and recall metrics as following:

$$Fscore = 2 \times \frac{precision \times recall}{precision + recall}$$

### Feature selection

Forward feature selection was applied to identify discriminative features for each term. Features were initially sorted based on KS test *P*-value comparing positive and negative samples. Sorted features were added sequentially to the model, and only features that improved the model performance based on *F*-Score were retained. No more than 100 features were allowed per model.

Once parameters are optimised, an additional cross-validation step may be applied to minimise potential bias to the initial split in training and testing gene sets. Although KCML predictions are robust in general to initial seed where on average 81% of the predictions were consistent with the initial run. SVM can be retrained based on the selected features and SVM parameters while using 10-fold cross-validation using all the annotated genes. Then, predictions that are consistent across 7/10 of the trained classifiers can be considered. Therefore, cross-validation or even leave-one-out can be used to obtain more stringent predictions following parameter optimisation.

Restricting KCML to gene annotations based on experimental evidence codes in UniProt (EXP, IDA, IEP, IMP, IGI and IPI) was tested but did not improve the performance. We also tested excluding genes in terms that are semantically similar to the term under classification from the negative class. This also did not have a significant effect on KCML performance as selecting these genes had a very low probability when a random sample from all genes was selected.

### Scoring metrics and selection of classifiable GO terms

The classification performance can be affected by multiple factors: (i) The quality of GO annotations. (ii) For a given GO term, not all annotated genes would perform that function in the investigated cell type or context. (iii) Incomplete data: the set of phenotypic readouts acquired by a certain experiment provides a partial description of cellular states. (iv) Perturbations efficiency.

AUROC provides a good indication on the true-positive rate versus false-positive rate of each GO term classifier at different SVM confidence cut-offs. However, AUROC is not sufficient, as some models will have very high false-positive rate that is balanced by high true-positive rate. Instead, we selected models based on combination of metrics to ensure the reliability of the model. Our cut-off for false-positive rate is < 20%, recall on test samples > 30% and recall on training samples > 40% as all classifiers of random gene sets scored less than this cut-off.

Random gene sets: 100 random gene sets were drawn with each sample having 200 genes. Then, KCML was trained to classify these random samples.

### Performance of clustering-based method

Principal component analysis was applied to reduce the dimensionality of the data. K-means and self-organising maps clustering methods were applied to the first 100 principal components which captured 96.25% of the variability in the data. The number of clusters was varied between 10 and 200 clusters, as it is not trivial to determine the number of clusters. Hierarchical clustering was also tested but found to result in one big cluster and many clusters with very few genes. Once clusters were determined, the overlap between genes in each of the clusters and genes in the 1,019 tested GO terms were used to compute the recall and AUROC.

### Performance of correlation-based methods

Pearson correlation coefficient between genes annotated with the same GO term was measured. A gene that has > 0.9 correlation with any of the other genes in a given GO term was counted as a true-positive if it is also annotated with that term or false-positive if it is not annotated with that term.

### Generation of GO term network for Image-based dataset

The classifiable GO terms based on the image-based dataset were connected by an edge if their overlap is > 0.70. GO terms that did not score an overlap above this threshold were connected to the term with the highest overlap. This was performed iteratively until all the terms got connected to the network. The generated network was visualised in Cytoscape v3.3.

### Interaction enrichment

For a given GO term classifier, we counted the number of positive and negative genes on one hand, and those that are first neighbours of the annotated genes for that term or not. Then, hypergeometric test was used to calculate the significance of the number of first neighbour interactions.

### Detection of seed effects

A pool of four siRNA was used in the image-based single-cell-resolved dataset. The siRNA seed was defined as the $2^{nd}$-$7^{th}$ position of the siRNA sequence. 2,616 out of 3,575 seeds occurred in four or more genes. On average, every seed occurred in 20.17 genes. If a seed associated with different genes was significantly enriched in the predicted genes list for a given GO term (Fisher's test right-tail, $P < 0.01$); then, all the predictions linking the genes targeted by this seed to the enriched term were considered off-target effects and filtered out (Fig EV5D).

### Subphenotypic space embedding

To generate the subphenotypic space that captures SVM decision boundaries, we applied logistic regression to learn feature weights for the selected features by SVM. We trained a logistic regression model to learn the predictions of a given SVM term classifier based on the selected features. Then, t-SNE embedding was generated based on the selected features by SVM term classifier multiplied by their corresponding weights.

### TCGA analysis

Silhouette clustering index of colorectal cancer molecular subtypes was used to determine clustering quality (separability and coherence of clusters) based on all genes, top predicted MSD genes or known MSD genes. Principal component analysis was used to reduce the number of dimensions. The first 2 principal components were used to compute the silhouette index of colorectal cancer molecular subtypes. The same number of principal components was used when considering the top 300 predicted MSD genes based on SVM confidence. To estimate the significance of MSD genes in the classification of colorectal cancer molecular subtypes, we generated a hundred random samples of 300 genes and computed the mean of average silhouette values for the hundred samples (t-test).

### Transcription-based signatures

MSD signatures were computed as the average expression of top predicted MSD genes (rank 1–300, rank 301–600, 301–1000) and known MSD genes. Olfactory receptor signature was computed as the average expression of olfactory receptors associated with MSD. WNT signature was computed as the average of the following WNT genes: WNT1, WNT2, WNT2B, WNT3, WNT3A, WNT4, WNT5A, WNT5B, WNT6, WNT7A, WNT7B, WNT8A, WNT8B, WNT9A, WNT10A and WNT10B. TGFβ signature was computed as the average of genes identified in Calon *et al* (2012), except for endothelial-associated genes.

Expression state of olfactory receptors was empirically determined depending on the distribution of their expression values as following: OR51B4 cut-off = 0.25, OR5K1 cut-off = 0.01 and olfactory receptor metagene cut-off = 0.1. Olfactory receptor metagene was defined by aggregating the expression state of many olfactory receptors as following: Each olfactory receptor was added iteratively to the metagene and retained if the metagene scored significant based on Kaplan Meier log-rank test (Fig EV7G). To avoid overfitting, a leave-one-patient-out cross-validation was performed and the gene is retained only if it scored significant in more than 85% of the tests. Forty-five olfactory receptors were selected out of 83 genes. To investigate potential spurious association between gene expression and patient survival, 100 random genes were drawn. No significant association with patient outcomes using Kaplan–Meier survival analysis was observed with $P < 0.005$.

Kaplan–Meier survival analysis and Cox proportional hazard modelling were performed in R.

### Independence of olfactory receptor metagene expression from neighbouring oncogenes amplification

Known cancer genes (Repana *et al*, 2019) that are located on the same cytogenetic bands as the olfactory receptors in our metagene were retrieved. To test whether the olfactory receptor metagene can predict survival independent of the expression of neighbouring cancer genes Cox proportional hazard test was performed (Table EV9).

## Data availability

Image-based single-cell data and KCML predictions are available on FigShare repository (Data ref: Sailem *et al*, 2020 and Data ref: Green & Pelkmans, 2020). Code is available on GitHub repository: https://github.com/hsailem/KCML and is also provided as Code EV1.

Expanded View for this article is available online.

### Acknowledgements

We thank Dr. Victoria Green and Sophie Tritschler for generating the image-based dataset which provides a great resource for biological discovery. We also thank members of Pelkmans group, Christine Orengo and Jon Lees (UCL), Andrew Zisserman (University of Oxford) for useful discussions. HZS is a Sir Henry Wellcome Fellow (Grant number: 204724/Z/16/Z).

### Author contributions

HZS conceived the work, designed and implemented KCML, performed all data analyses and wrote the paper. LP provided the image-based dataset and thoroughly discussed the results with HZS. JR and LP reviewed and edited the paper.

### Conflict of interest

The authors declare that they have no conflict of interest.

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
