## [Review Process File · Molecular Systems Biology]

KCML: a machine-learning framework for inference of multi-scale gene functions from genetic screens

Heba Sailem, Jens Rittscher and Lucas Pelkmans.

Review timeline:

Submission date:	27 th June 2019
Editorial Decision:	5 th August 2019
Appeal received:	23 rd October 2019
Editorial Decision:	20 th December 2019
Revision received:	1 st February 2020
Accepted:	6 th February 2020

Editor: Maria Polychronidou

Transaction Report:

1st Editorial Decision

5th August 2019

Thank you again for submitting your work to Molecular Systems Biology. We have now heard back from the two referees who agreed to evaluate your manuscript. As you will see below, the reviewers raise substantial concerns on your work, which unfortunately preclude its publication in Molecular Systems Biology.

Overall, the reviewers appreciate that developing methods for inferring gene function from large-scale screens is a relevant topic. However, reviewer #2 is not convinced that the performed analyses suffice to robustly support that KDML is a decisive and broadly applicable methodological advance and points out that its ability to aid new biological discoveries remains to be further demonstrated. Moreover, both reviewers emphasize that the absence of code and the lack of a detailed documentation of the methodology hamper the thorough assessment of the study in its present form. As such, both reviewers indicated that they do not support publication in Molecular Systems Biology.

Taken together, and considering the overall rather low level of enthusiasm expressed by the reviewers I am afraid I see no other choice than to return the manuscript with the message that we cannot offer to publish it.

REFeree REPORTS

Reviewer #1:

Sailem et al propose KDML, a method to infer gene functions from screening data. The method is based on building classifiers for individual gene ontologies. Then they used this GO term specific models to predict new genes belonging to the same GO term. KDML is then applied to 3 studies: with gene expression, cell viability and image features. The method is validated on analysis of

TCGA data.

Extracting biological knowledge from screenings is an important branch of computational systems biology, and KDML is an interesting addition to this area.

We have the following questions and comments:

1. Any paper should provide code for reproducibility. Even more a paper presenting a method must provide the code - is not possible to truly assess the method, and the method will not be used by others. 'Code available upon request' does not seem appropriate.

1.1. Not as critical but even better would be if the authors would use free software to implement KDML - the analyses are standard machine learning / statistics that can be implemented e.g. in R or Python easily. This way the use would be broader.

2. A general comment is that the paper describes succinctly the method, and then in large detail the applications and validations. For a method paper, we would have expected more information on the methods. Lacking this, there are a number of questions we outline below, that we could not get from reading the paper:

2.1. In the scoring is not clear always what are the Authors use F-score to assess the models. They also provide in some cases REcall, or an AUC. The AUC is not clear if it is AUROC, AUPROC or something else, and how it is built (e.g. which parameter is changed to obtain the different points in the curve). In any case, it is informative to show the actual curves. For unbalanced data sets single numbers (e.g. Recall or AUROC) can be misleading.

2.1.1. In particular, recall in Figure 1, is it calculated for >50% prediction probability? Why don't they use Precision-Recall Curve AUCs?

2.2. KDLM was benchmarked against random gene sets and clustering methods. We do not see this as a fair or relevant comparison, can authors elaborate on this? And how can they use clustering methods to infer gene ontology, or any other classification that can be compared against KDLM?

2.3. Authors use 70 % for training and 30 % of genes for testing. How do results change if the split in training and test change?

3. Applications:

3.1. It has been observed that there are many spurious associations of expression and survival.

Authors should compare their results on the TCGA at least against randomly chosen genes at least.

3.2. For MSD signature in TCGA, did they used only predicted genes, or predicted + already known genes? If the latter, how would Fig6C-E change, when they would use only the predicted and only the known genes?

4. For Olfactory receptor metagene, they selected the genes using an iterative process (Fig. EV6E). Can not this lead to overfit?

Minor:

5. In Fig6H-J probably figure legend is wrong (Grada 3+ is always, should not be Grade 3+ and also 3-?)

6. Figure 5 could be supplementary, does not seem so informative (as opposed to the very dense Figure 1)

Reviewer #2:

The function of a large portion of human genes remains unknown although many high throughput gene perturbation screening have been performed. Furthermore, the utility of functional annotation databases such as gene ontology is limited by cell type and experimental context, though to what degree is unknown. To resolve these problems and expand limitation of gene ontology annotation, the authors suggested a machine learning approach, KDML, to connect gene ontology annotation and genes learned from pre-existing high throughput screening data, such as image based screens and viability screens. In this paper, they identified olfactory genes and TNFbeta and WNT signaling pathway genes are enriched in predicted genes of mesoderm development from image based screens

using HCT1116 cell line, and "validated" that mesoderm development genes and olfactory, TNF beta and WNT signaling pathways are correlated in TCGA colon cancer data. However, since their validations are limited to one analysis case, it's not clear this algorithm can be generalized to other data. While intriguing in concept, the manuscript as described has major flaws, both technical and logical, and should not be published as is.

First, the method is poorly described. The manuscript as a whole uses highly fluent English that unfortunately frequently fails to succinctly describe the concepts and the results that it reports. A much clearer description of the algorithm's steps is warranted - in plain, not algorithmic/mathematical, language. It is exceedingly difficult to decipher what exactly the KDML algorithm is doing here. Walking through an example might be informative.

Second, the entire manuscript appears to be constructed around some foundational logical flaws. Biology at all scales, but especially at the cell biological scales examined here, is inherently modular: groups of genes operate together to perform a bioprocess (often a protein complex or enzymatic or signal transduction pathway). These modules are repurposed to operate at various stages of the organismal/developmental and cellular life cycles. The TGF-beta signaling pathway is associated with epithelial to mesenchymal transition, and WNT signaling is involved in basal-apical polarity in epithelial sheets. That these genes are associated with mesodermal development in one context and EMT in another is neither surprising nor novel, and for the authors to assert that "MSD (mesodermal development) genes might play a role in colorectal cancer..." (p11) either fails to acknowledge this underlying modularity - of course genes annotated as MSD are the same ones annotated as enriched in colorectal cancer; WNT pathway mutations are a hallmark of colorectal cancer - or fails to cleanly articulate what novelty the algorithm is bringing to the table.

Likewise, the discussion of olfactory receptor involvement in any bioprocess is a minefield, for several reasons. Olfactory receptors show high familial sequence similarity and are therefore difficult to discriminate in, e.g., RNAseq gene expression studies (since reads map to multiple ORs). Differential expression in cancer is also confounded by copy number amplifications of nearby oncogenes and any novel association between tumor gene expression and clinical outcome should be corrected for these known issues. Simply put, extraordinary claims require extraordinary evidence, and associating cancer clinical relevance with ORs is an extraordinary claim.

Other specific issues:

- This study proposed an algorithm for processing three different data types from screens. But, "validation" of predictions are mostly described for the image based screen. It's better to validate the results from other data in detail.

- The performance of unsupervised algorithms, K-means and SOM, are unexpectedly low. It should be compared with potential supervised algorithms. For example, in the case of the viability screen, a functional relationship between two genes can be predicted by calculating co-dependency of two genes across several cell lines. Also, gene ontology annotations can be expanded by finding genes having high co-dependency with the annotation.

- Authors used a collapsed set of 100-500 gene ontology terms for train and test. However, the number of predicted gene list per term regarding Fig 2D exceeds 1,000 genes and even close to 3,000 genes. The authors should explain this discrepancy and provide rationale for such large gene sets.

- The effect of a gene in a biological process can be bidirectional, positive or negative. However, it's not clear that KDML can account these bidirectional effects of genes in single GO term.

- In Fig 6C, PCA plots using all genes and top 300 predicted MSD genes were compared. How about comparing top 300 genes and rank 301--600 or 301-1000 genes? Is there significant difference between them?

- Multiple hypothesis corrected p-value for co-expression is better to describe significance.

Minor points

- There is no explanation of what is HT-GPS in introduction. Is it a popular term?
- In page 8, multicellular organismal singling function -> signaling function
- In page 9, pvalue <4.9-139 -> pvalue < 4.9e-139
- In Fig 6H,6I, and 6J, all labels have same annotation 'Grade 3+', also green and blue as well.

Author's Appeal

23rd October 2019

Letter to the Editor:

Your comments and the reviewers' feedback have been very helpful. We believe our initial submission did not fully convey the power of KDML and its potential for discovering novel gene functions. The approach is, as reviewer 2 noted, intriguing. To highlight the principal novelty, utility, robustness, and impact of the approach we present you with an updated version of the paper which has been substantially revised:

- **Novelty.** To our knowledge this is the first systematic attempt to link gene ontology terms with gene perturbation and imaging data. We believe that this approach has far reaching consequences for the entire field. Our machine learning-based approach provides a directly implementable concrete solution for discovering novel gene functions from different types of large-scale perturbation data.
- **Utility.** Currently, less than 2% of the data generated by image-based genetic screens are utilised for inference of gene function. By applying KDML to studies which investigate phenotypic changes on different scales, we demonstrate that this approach is generalisable to vastly different datasets. The code will be made available on Github upon publication.
- **Robustness.** Recently published UniProt annotations allow us to investigate the robustness of our approach. We show that KDML can predict a significant number of new UniProt annotations that were never seen by our method, based on datasets utilising different perturbation technologies and readouts. This not only validates KDML predictions but also illustrates its generalizability.
- **Impact.** To address the reviewer comments, we have substantially revised the presentation of our biological findings. However, we would like to stress that these provide compelling examples of discoveries that could be enabled by KDML.

We would kindly ask you to consider the revised version of our paper. KDML provides an important advancement to system genetics and we believe that this work will be of great interest to the readership of Molecular Systems Biology.

Please note that a preprint of the original submitted manuscript is available under the

Response to the comments of Reviewer #1

“Sailem et al propose KDML, a method to infer gene functions from screening data. The method is based on building classifiers for individual gene ontologies. Then they used this GO term specific models to predict new genes belonging to the same GO term. KDML is then applied to 3 studies: with gene expression, cell viability and image features. The method is validated on analysis of TCGA data.

Extracting biological knowledge from screenings is an important branch of computational systems biology, and KDML is an interesting addition to this area.”

We thank the reviewer for highlighting the importance of approaches like KDML as it facilitates biological discovery from screening data. Despite their prevalence, screening data are the least utilised datasets in the field of systems biology. Novel computational approaches, such as KDML, are critical to extract value from these studies, which can be extremely costly and labour intensive.

“1. Any paper should provide code for reproducibility. Even more a paper presenting a method must provide the code - is not possible to truly assess the method, and the method will not be used by others. 'Code available upon request' does not seem appropriate.”

We agree with the reviewer and now share our code as supplementary software, which will be made a publicly accessible Github repository upon acceptance of this manuscript.

“1.1. Not as critical but even better would be if the authors would use free software to implement KDML - the analyses are standard machine learning / statistics that can be implemented e.g. in R or Python easily. This way the use would be broader.”

We will release a compiled version of the code upon acceptance so the user can run it without having MatLab installed.

“2. A general comment is that the paper describes succinctly the method, and then in large detail the applications and validations. For a method paper, we would have expected more information on the methods. Lacking this, there are a number of questions we outline below, that we could not get from reading the paper:”

We apologise that the methodology of KDML was not fully clear to the reviewer. We now extensively revise the text and Figure 1 to ensure the clarity of KDML methodology.

“2.1. In the scoring is not clear always what are the Authors use F-score to assess the models. They also provide in some cases REcall, or an AUC. The AUC is not clear if it is AUROC, AUPROC or something else, and how it is built (e.g. which parameter is changed to obtain the different points in the curve)... For unbalanced data sets single numbers (e.g. Recall or AUROC) can be misleading”

This is an important point and we now discuss the metrics utilised in more detail.

As the reviewer suggested using one metric can be misleading. It is possible that a high AUROC still yields an unacceptable number of false predictions for the purpose for predicting gene functions. Therefore, we use multiple

measures to assess the results. Classifiable terms are selected based on the recall (sensitivity) as well as precision (specificity) which we find to also yield good AUROC. F-score was used for parameter optimisation, as AUROC can be computationally intensive.

“In any case, it is informative to show the actual curves.”

We now show the actual curves for the three datasets (Fig. EV1E).

“2.2. KDLM was benchmarked against random gene sets and clustering methods. We do not see this as a fair or relevant comparison, can authors elaborate on this? And how can they use clustering methods to infer gene ontology, or any other classification that can be compared against KDLM?”

Indeed, currently there is no other method that offers such systematic analysis of high-throughput gene perturbation screen (HT-GPS) datasets based on gene ontology or other functional annotations. Unsupervised clustering approaches are the most commonly used, followed by functional enrichment for GO terms. By comparison, the resulting phenotypic clusters perform much worse than KDML in detecting functional phenotypes.

In accordance with Reviewer 2 suggestion we also compare KDML results to enrichment based on high correlation between gene pairs within the same GO term which show much poorer performance (Fig. 2D).

“2.3. Authors use 70 % for training and 30 % of genes for testing. How do results change if the split in training and test change?”

We thank the reviewer for raising this important point. To test for this, we follow our training by additional 10-fold cross validation over all samples. Then we consider predictions that are predicted as positive based on 7/10 of the trained classifiers. We found that KDML predictions are robust in general where on average 81% of the predictions will be consistent with our initial run. Therefore, 10-fold cross validation might be used to obtain more stringent predictions following parameter optimisation as we explain now in the Methods section.

“3. Applications:

3.1. It has been observed that there are many spurious associations of expression and survival. Authors should compare their results on the TCGA at least against randomly chosen genes at least.”

We now compare against 100 randomly selected genes with predicted mesoderm and known cancer genes excluded (Repana et al, 2019). Random genes did not significantly predict survival as explained in the text.

“3.2. For MSD signature in TCGA, did they used only predicted genes, or predicted + already known genes? If the latter, how would Fig6C-E change, when they would use only the predicted and only the known genes?”

We used only predicted mesoderm genes. We now compare the performance of predicted genes versus known mesoderm genes in Fig. 6C which demonstrates that they result in consistent signatures.

“4. For Olfactory receptor metagene, they selected the genes using an iterative process (Fig. EV6E). Can not this lead to overfit?”

To avoid potential overfitting, we now perform ‘leave one patient out’ for each tested olfactory receptor. Then only olfactory receptors that were significant in more than 85% of the folds were considered in the final metagene (Fig. 6I,K). Based on this, we did not observe substantial change in the selected genes.

“Minor: 5. In Fig6H-J probably figure legend is wrong (Grada 3+ is always, should not be Grade 3+ and also 3-?)”

Corrected.

“6. Figure 5 could be supplementary, does not seem so informative (as opposed to the very dense Figure 1)”

We moved old Fig. 5 to the supplementary and expanded Fig. 1 into two figures.

Response to the comments of Reviewer #2

“The function of a large portion of human genes remains unknown although many high throughput gene perturbation screening have been performed. Furthermore, the utility of functional annotation databases such as gene ontology is limited by cell type and experimental context, though to what degree is unknown. To resolve these problems and expend limitation of gene ontology annotation, the authors suggested a machine learning approach, KDML, to connect gene ontology annotation and genes learned from pre-existing high throughput screening data, such as image based screens and viability screens. In this paper, they identified olfactory genes and TNFbeta and WNT signaling pathway genes are enriched in predicted genes of mesoderm development from image based screens using HCT1116 cell line, and “validated” that mesoderm development genes and olfactory, TNF beta and WNT signaling pathways are correlated in TCGA colon cancer data.

However, since their validations are limited to one analysis case, it's not clear this algorithm can be generalized to other data.”

This is a valid concern, which we addressed by validating the results from different datasets in an unbiased manner. New annotations (July 2019) that have been added by UniProt after our initial download (Mar 2018) are now used to show that KDML is capable of predicting many of the new annotations from different types of multivariate gene perturbation datasets. These results illustrate the power and generalisability of KDML (Fig. 2E-F).

“While intriguing in concept, the manuscript as described has major flaws, both technical and logical, and should not be published as is.”

We thank the reviewer for finding our study intriguing and regret that the initial presentation of our work gave the impression that the study has major flaws. The logic and technical setup of KDML is inspired by hallmark studies in the protein function prediction field, where protein function is predicted from its structure or phylogeny (Jiang et al, 2016; Dey et al, 2015; Radivojac et al, 2013). KDML is the first methodology to adapt this proven concept to analyse genetic screens where phenotypic similarity has been shown to reflect functional relationships. We believe that KDML is an important contribution to the field especially since in most studies less than 2% of the information from screening datasets is utilised for functional inference.

“First, the method is poorly described. The manuscript as a whole uses highly fluent English that unfortunately frequently fails to succinctly describe the concepts and the results that it reports. A much clearer description of the algorithm's steps is warranted - in plain, not algorithmic/mathematical, language. It is exceedingly difficult to decipher what exactly the KDML algorithm is doing here. Walking through an example might be informative.”

We regret that the reviewer did not find some of the results in the manuscript clear and we have now significantly revised the manuscript and the figures to ensure the clarity of KDML methodology.

“Second, the entire manuscript appears to be constructed around some foundational logical flaws. Biology at all scales, but especially at the cell biological scales examined here, is inherently modular: groups of genes operate together to perform a bioprocess (often a protein complex or enzymatic or signal transduction pathway). These modules are repurposed to operate at various stages of the organismal/developmental and cellular life cycles.”

We believe that the logic behind KDML was misunderstood by the reviewer as it indeed supports the well-known modularity of gene programs and their combinatorial use in different biological contexts. The novelty lies in the fact that KDML enables the computational inference of such modular gene programs through automated analysis of multivariate quantitative data from perturbation screens. This allows the systematic identification of, for instance, multiple signalling pathways that result in similar quantitative phenotypes. We further emphasise this in our discussion;

“... These results exemplify how KDML can be used to automatically interrogate quantitative phenotypic profiles to identify combinatorial use of modular gene programmes in different contexts.”

“The TGF-beta signaling pathway is associated with epithelial to mesenchymal transition, and WNT signaling is involved in basal-apical polarity in epithelial sheets. That these genes are associated with mesodermal development in one context and EMT in another is neither surprising nor novel, and for the authors

to assert that "MSD (mesodermal development) genes might play a role in colorectal cancer..." (p11) either fails to acknowledge this underlying modularity - of course genes annotated as MSD are the same ones annotated as enriched in colorectal cancer; WNT pathway mutations are a hallmark of colorectal cancer - or fails to clearly articulate what novelty the algorithm is bringing to the table."

We appreciate the reviewer's comment and it was not our intention to claim that these observations are, by themselves, novel. The novelty here is our algorithmic method for analysing HT-GPS data, which can, for the first time, automatically infer modular gene programs such as those involving WNT, TGFbeta, cell adhesion and cytoskeleton, and link them to potential biological functions. This proves that KDML can discover biologically relevant information as the reviewer articulated very clearly. Although these predictions are not *persé* surprising, the current GO annotations did not include many of the genes in these pathways (Fig. 6A). Furthermore, to our knowledge, no other study showed that perturbation of these genes result in cell clumping in HCT116.

"Likewise, the discussion of olfactory receptor involvement in any bioprocess is a minefield, for several reasons. Olfactory receptors show high familial sequence similarity and are therefore difficult to discriminate in, e.g., RNAseq gene expression studies (since reads map to multiple ORs)."

We appreciate the concern raised by the reviewer, however such effects would be consistent across patients with low or high expression levels of olfactory receptor genes and therefore still support the correlation between olfactory receptors expression and patient outcomes. To further assure the reviewer, we identified genes with highest sequence similarity to the olfactory receptors in our metagene (Olender et al, 2013). We found that, in general, these genes have moderate similarity with other olfactory receptors and therefore they are not likely to be misaligned (Table EV5 and Fig. EV7I). Rigorous estimation of the probability of read misalignment of these genes is beyond the scope of this work.

"Differential expression in cancer is also confounded by copy number amplifications of nearby oncogenes and any novel association between tumor gene expression and clinical outcome should be corrected for these known issues."

We now show that our olfactory receptor metagene can predict survival independent of the expression of neighbouring oncogenes (Table EV9). As amplification of neighbouring oncogenes should be reflected in their expression levels, this analysis confirms the predictability of olfactory receptors of clinical outcomes. As explained in the methods this analysis is performed as following:

Independence of olfactory receptor metagene expression from neighbouring oncogenes amplification: Known cancer genes (Repana *et al*, 2019) that are located on the same cytogenetic bands as the olfactory receptors in our metagene were retrieved. To test whether the olfactory receptor metagene can predict survival independent of the expression of neighbouring cancer genes, cox proportional hazard test was performed (Table EV9).

“Simply put, extraordinary claims require extraordinary evidence, and associating cancer clinical relevance with ORs is an extraordinary claim.”

The presentation of this claim has been substantially revised to clarify that KDML is used here to generate hypotheses that need further empirical validation. But there is increasing evidence for an association between olfactory receptors and cancer as well as other diseases (see for a review Lee *et al.*, Nature Reviews Drug Discovery, 2019). Our result is in concordance with these studies as it indicates a significant association between the expression of many olfactory receptors based on OR metagene and patient survival, suggesting that they can be used as a potential biomarker. However, further validation of their clinical relevance is beyond the scope of this methodological paper.

“This study proposed an algorithm for processing three different data types from screens. But, “validation” of predictions are mostly described for the image based screen. It’s better to validate the results from other data in detail.”

Validation examples from the three datasets are now added as explained above. Nonetheless, we note that imaging data are the most challenging to analyse and therefore the most underutilised. Moreover, imaging provides structural and spatial information with single-cell resolution that are not offered by other methods.

“The performance of unsupervised algorithms, K-means and SOM, are unexpectedly low. It should be compared with potential supervised algorithms. For example, in the case of the viability screen, a functional relationship between two genes can be predicted by calculating co-dependency of two genes across several cell lines. Also, gene ontology annotations can be expanded by finding genes having high co-dependency with the annotation.”

To our knowledge, there is no other supervised method for inference of gene functions from genetic perturbation screens. We perform the analysis requested by the reviewer. The number of highly correlated genes within one GO term (Pearson Correlation > 0.9) represents a very small fraction of annotated GO terms (Fig. 1D).

“Authors used a collapsed set of 100-500 gene ontology terms for train and test. However, the number of predicted gene list per term regarding Fig 2D exceeds 1,000 genes and even close to 3,000 genes. The authors should explain this discrepancy and provide rational for such large gene sets.”

We thank the reviewer for highlighting this confusion. KDML was trained on GO terms that have between 100-500 annotated genes. While in the aforementioned figure (now Fig. 3D) the indicated number is based on the number of predicted genes by KDML. We now explain this clearly in Fig. 3D. Potential reasons for the high number of false positive predictions are thoroughly discussed in the second paragraph of the discussion. As we note in the manuscript these predictions can be weighted based on SVM rank (phenotypic similarity) or other biological databases such as protein-protein

interactions or KEGG pathways.

“The effect of a gene in a biological process can be bidirectional, positive or negative. However, it's not clear that KDML can account these bidirectional effects of genes in single GO term.”

Currently, KDML can account for bi-directionality of gene function only if positive and negative regulators are included as different GO terms. However if these are in a single term then KDML will search for the most consistent perturbation phenotype across the genes in that term. This issue would be the same for any method using functional enrichment approaches including typical clustering-based methods.

“In Fig 6C, PCA plots using all genes and top 300 predicted MSD genes were compared. How about comparing top 300 genes and rank 301--600 or 301-1000 genes? Is there significant difference between them?”

Similar results are obtained when investigating predicted mesoderm genes with rank 301-600 or 301-1000 as shown in Fig. 6 D.

“Multiple hypothesis corrected p-value for co-expression is better to describe significance.”

Unfortunately, it is not clear which specific results are being referred to, as we only compared average expression of mesoderm genes against average signatures of TGFbeta and WNT genes.

“Minor points

- There is no explanation of what is HT-GPS in introduction. Is it a popular term?”

As explained in the introduction HT-GPS stand for High Throughput Genetic Perturbation Screens

“- In page 8, multicellular organismal singling function -> signaling function

- In page 9, pvalue <4.9-139 -> pvalue < 4.9e-139

- In Fig 6H,6I, and 6J, all labels have same annotation 'Grade 3+', also green and blue as well.”

These and other typographical errors have been corrected.

2nd Editorial Decision

20th December 2019

Thank you again for submitting your work to Molecular Systems Biology. We sent the study to the same reviewers who evaluated your previous submission. We have now heard back from one of the two referees. If we receive comments from reviewer #2 within the next few days, I will forward them to you. As you will see below, reviewer #1 thinks that the study has improved as a result of the performed revisions. They raise however a few remaining concerns, which we would ask you to address in a revision.

REFeree REPORTS

Reviewer #1:

I thank the authors for their efforts to address my comments. These have been largely fulfilled.

It would have helped that the resubmitted version marked the new content in a different color, as is customary.

I commend them for making their code available - even if it is a pity is based on commercial software for which I have no license, and hence I can not test it.

By the way, I could not find a license in the code, authors should add one.

2 points remain unsolved:

1#

Based on my comment:

3.1. It has been observed that there are many spurious associations of expression and survival.

Authors should compare their results on the TCGA at least against randomly chosen genes at least."

Authors state:

We now compare against 100 randomly selected genes with predicted mesoderm and known cancer genes excluded (Repana et al, 2019). Random genes did not significantly predict survival as explained in the text.

Why removing the known genes? I think this biases the analysis. If I do a random search, I search blindly without removing genes. Also, they do not remove for the DKML search known genes, do they? That does not seem a fair comparison.

2 #Authors now include AUROC, though in my comment I had pointed out that AUPR are likely more adequate due to unbalanced positives/negatives distributions. They should report AUPR (or both).

Reviewer #2:

My congratulations to the authors on this substantially revised and improved manuscript. The clarity of the presentation of the central ideas behind the KDML classifier are vastly improved over the initial submission, and the results/validation do not overreach with the conclusions.

Also, making the software is available for review is another critical requirement that the authors have met. Unfortunately, it doesn't execute properly, most likely due to file path navigation/syntax issues. I got the following error when executing main.m (Matlab 2019b, Linux):

2nd Revision - authors' response

1st February 2020

Remarks to the reviewers

Reviewer 1

"I thank the authors for their efforts to address my comments. These have been largely fulfilled"

We thank the reviewer for their support of our work.

“I commend them for making their code available - even if it is a pity is based on commercial software for which I have no license, and hence I can not test it.”

The code is now available on GitHub (<https://github.com/hsailem/KCML>). MATLAB is a widely used software, but we also release a compiled version that can be used by installing MATLAB run time which is free to install.

“By the way, I could not find a license in the code, authors should add one”

We added a standard open source license file to KCML package.

“Why removing the known genes? I think this biases the analysis. If I do a random search, I search blindly without removing genes. Also, they do not remove for the KDML search known genes, do they? That does not seem a fair comparison.”

We now repeated the analysis while including all genes. We do not observe a significant correlation at $p\text{-value} < 0.005$.

“Authors now include AUROC, though in my comment I had pointed out that AUPR are likely more adequate due to unbalanced positives/negatives distributions. They should report AUPR (or both).”

We thank the reviewer for his suggestion. We now report AUPR (Fig. EV1H) and describe the results in the second paragraph on page 9. We would like to emphasize that our approach is weakly supervised as we are using gene ontology annotations which only provide noisy labels. This can result in low precision values. Combining KCML predictions with other information such as protein-protein interactions or enrichment for certain signaling pathways can be useful in further prioritising potential follow-up genes as discussed in the second paragraph on page 14. We note that our method brings a significant improvement to existing methods.

Reviewer 2

“My congratulations to the authors on this substantially revised and improved manuscript. The clarity of the presentation of the central ideas behind the KDML classifier are vastly improved over the initial submission, and the results/validation do not overreach with the conclusions. Also, making the software is available for review is another critical requirement that the authors have met.”

We are glad that the message and the method presented in our manuscript is now much clearer to the reviewer.

“Unfortunately it doesn't execute properly, most likely due to file path navigation/syntax issues. I got the following error when executing main.m (Matlab 2019b, Linux):”

We now tested our code on Linux, Window, and Mac. The user can specify the directory names of their data in the configuration file as described in KCML package.

Accepted

6th February 2020

Thank you again for sending us your revised manuscript. We are now satisfied with the modifications made and I am pleased to inform you that your paper has been accepted for publication.

Corresponding Author Name: Heba Sailem

Manuscript Number: 19-9083R-Q